# Links between Daytime Napping, Night-Time Sleep Quality and Infant Attention: An Eye-Tracking, Actigraphy and Parent-Report Study

**DOI:** 10.3390/children9111613

**Published:** 2022-10-23

**Authors:** Nabil Hasshim, Jessica Bramham, Jennifer Keating, Rebecca A. Gaffney, Lisa Keenan, Sarah Conroy, Fiona McNicholas, Alan Carr, Michelle Downes

**Affiliations:** 1School of Psychology, University College Dublin, D04 V1W8 Dublin, Ireland; 2School of Applied Social Sciences, De Montfort University, Leicester LE1 9BH, UK; 3Department of Child & Adolescent Psychiatry, School of Medicine and Medical Science, University College Dublin, D04 V1W8 Dublin, Ireland

**Keywords:** sleep, infancy, attention development, eye tracking

## Abstract

The current study explored the potential influence of infant sleep, measured by parental report and actigraphy, and family functioning on attention development using eye tracking. The use of actigraphy in parallel with parental report, has the advantage of measuring participant’s sleep throughout the night without parental observation and the ability to objectively assess sleep quality. An eye-tracking version of the Gap-Overlap task was used to measure visual attention. Questionnaires and behavioural assessment were used to assess family function, and general cognitive development. Fifty infants (*Mean age* = 13.44 months, *SD* = 3.10) participated in the study, 23 of which had full final datasets. Results show that daytime sleep duration, as measured by parental report, and proportion of light sleep at night, as measured by actigraphy, are linked to visual attention. A higher proportion of light sleep, a marker of poorer sleep quality, and less daytime sleep were negatively linked with facilitation and disengagement on the Gap-Overlap task. Family functioning was not associated with attention. The results provide initial evidence that in addition to the amount of daytime sleep; quality of night-time sleep as measured by proportion of light sleep, is a potentially useful sleep variable which requires further focus in the study of attention development.

## 1. Introduction

The role of sleep in cognitive functioning has been well documented, with a steadily growing interest in the important role of sleep for the developing child. However, due to the methodological challenges in obtaining objective data, such as using polysomnography (PSG), researchers typically rely solely on parent-reports of infants’ sleep. Although estimates of children’s sleep habits, such as bedtime and sleep duration, is easily obtained via such parental report, they do not provide an accurate measure of sleep quality with more recent research suggesting that factors such as age of infant and temperament of both infant and parent, can play a role in their accuracy [1,2,3,4,5]. The increased use of actigraphy in recent research has allowed for a more detailed study of sleep, specifically in the factors indicating its quality and not just quantity, and the current study aims to further contribute to the literature of this nascent field [6,7]. Actigraphy quantifies the amplitude of movement during sleep and one advantage is that it proves several different measures of sleep quantity and quality.

### 1.1. Sleep and Attention

Sleep disturbances are commonly reported in children with known attention deficits such as attention deficit hyperactivity disorder (ADHD) and are more generally associated with poorer child and family outcomes, including attention problems [8,9,10,11]. The body of existing developmental research that explores sleep and attention has mainly focused on this relation in the context of ADHD or sleep disorders, such as obstructive sleep apnoea, in school age children [12,13]. It has been reported that approximately one in four children with sleep problems in infancy will later meet diagnostic criteria for ADHD [14]. However, there is a paucity of studies on the influence of sleep on the development of concurrent attention skills in children younger than school age. There is a general prevalence of ADHD in 7.2% of children worldwide and recent research has shown that infant attention is linked to later ADHD symptomatology in children [15,16,17]. This is also despite a growing body of evidence for the influence of sleep in the first two years of life on attention skills later in childhood [18].

In the first study to objectively investigate the influence of sleep in infancy on childhood attention control using actigraphy, Sadeh and colleagues [19] found that lower quality sleep at 12 months (*N* = 43) predicted poorer attention regulation and more behavioural problems at preschool age. However, they did not assess concurrent attention in infancy. Poorer attention orienting and sleep actigraphy at 4 months in a group of preterm infants on a neonatal intensive care unit (*N* = 65) was related to higher levels of distractibility at 18 months [6]. In their study of night-time sleep quality at 12 months, Bernier and colleagues [20] found that parent-reported sleep was linked to executive functioning and working memory in early childhood. However, more recently Propper et al. [21] found no link between parent-reported sleep quality at 18 months and later teacher reported problems of attention in the early school years (though a link was observed for aggression).

Disparities in previous research on the links between sleep and attention in early development may be due to diverging methods of sleep and attention assessment with most studies reliant on parental report or behavioural observation of attention. Objective eye-tracking methodologies are less prone to human error and provide better spatial and temporal resolution than traditional behavioural methods [22]. In particular, eye-tracking methodology provides us with an opportunity to passively capture the development of infant attention as it offers a direct window into their ability to efficiently selectively attend to their surroundings in a non-biased way [23]. Downes and colleagues [24] recently used eye tracking methodologies to provide evidence for emerging attention differences by 12 months in very preterm infants. Perra and colleagues [25] are currently exploring the feasibility of using eye tracking methodologies for eye-tracking based cognitive training and assessment of attention in preterm infants. Although there has been an increased focus on the relation between sleep and attention in early development, to our knowledge there are no published studies have yet used objective measures in parallel with parent reported sleep to explore the link between sleep and concurrent attention control in infants. Additionally, a recent emphasis has been separately placed on the development of both infant sleep and attention in the context of extrinsic and intrinsic factors such as family environment [1,26,27]. There is a growing body of research that shows links between poorer sleep and poorer family functioning, as well as poorer attention and poorer family functioning in typically developing children and paediatric patient populations (e.g., [28,29]). Therefore, consideration of the potential role of external distal factors, such as family functioning, on the relation between sleep and attention in infancy is required.

### 1.2. Current Study

The current study assessed early attention control development using eye-tracking measures of visual attention and participants’ sleep using sleep diaries alongside actigraphy, an ankle worn device that detects movement. The use of sleep actigraphy in infant research is relatively new, and so there is a wide variation of sleep variables reported [30]. Different actigraphy measures have been used in different ways across studies that explore infant sleep development [4]. For example, Pisch et al. [7] reported duration of sleep at night, wake after sleep onset (WASO), and number of wake events. Konrad et al. [31] reported sleep efficiency and duration of daytime naps; while Scher [32] included latency between bedtime and falling asleep, average activity during sleep, and long wake periods. The use of different measures to indicate sleep quality across studies of cognitive development has led to uncertainty as to which ones may be the most appropriate in the context of the development of attention control. Furthermore, Camerota et al. [1] highlighted how researchers tend to cherry-pick sleep variables that are analysed and reported. As such, the current research explored all available variables provided by the actigraphy measure to report both those that do and do not impact attention. Finally, potential links between family function, attention and sleep were also explored.

The primary aim of the current study was to investigate the relation between day and night sleep and visual attention control in infants. Secondary aims included the exploration of potential links with available actigraphy measures to inform future investigations of infant sleep and attention (full list of measures below in *Actigraphy* section of the Methods,) and the exploration of the potential role of family function in the relation between sleep and attention. While it is was generally expected that infants with poorer night sleep quality and poorer daytime sleep would show poorer attention control on the lab-based tasks based on previous research [20], the study also aimed to explore available sleep variables generated by actigraphy to identify which variables are potentially most relevant for the future study of infant attention development using objective methods.

## 2. Materials and Methods

### 2.1. Participants

Fifty typically developing infants (24 female; age: *M* = 13.44 months, *SD* = 3.10) were recruited. Forty-four families identified as White Irish/British, two as mixed, and four of other ethnicities. The maternal education levels were as follows: 2 Leaving Cert/A-levels, 3 Level 6 Cert, 12 Undergraduate, 19 Postgraduate, and 16 indicated other education backgrounds or did not disclose. Due to participant non-compliance on certain tasks, not all participants were included in the analyses (see Table 1 for the breakdown of the number of participants in each measure). Participants had a mean developmental quotient of 102.5 (*SD* = 11.08) on the Bayley Scales of Infant Development [33] and family function score [34] of 1.87 (*SD* = 0.46).

### 2.2. Procedure

Ethical approval was obtained from University College Dublin Human Research Ethics Committee and parents provided informed consent before data collection began in the laboratory. During the testing session, all infants completed the general cognition domain of the Bayley Scales of Infant and Toddler Development, Third Edition (Bayley, 2006) and the eye-tracking tasks. At the end of the laboratory session, families received questionnaires, a sleep diary, and an actigraphy watch to record the infant’s sleep for the following week.

### 2.3. Measures

#### 2.3.1. Sleep Assessment

##### Sleep Diary

Parents received sleep diaries to record times the infants went to bed, got out of bed, night awakenings, and daytime sleep. Sleep diaries also recorded the time the infant was last fed before going to bed, whether they were put to bed awake/asleep, and whether it was a typical day. As well as augmenting actigraphy data, sleep diary entries were used to determine sleep during the day since some participants only wore the actigraph during night-time. *Day Sleep Time (DST)* in minutes, was calculated from the average daily nap times recorded by the parent.

##### Actigraphy

The use of actigraphy has been validated against PSG measures in their accuracy for identifying sleep and wake episodes in infants [30,35]. Participants received a micro-mini actigraph (Motionlogger Watch; Ambulatory Monitoring Inc., Ardsley, NY, USA) to track sleep-wake activities. Actigraphs recorded data at one-minute epochs in the zero-crossing mode. Parents were requested to attach the actigraph to the infant’s ankle for a period of at least one week. It was expected that infants might refuse to wear the actigraph during the day, parents were assured that this was fine and so actigraphy data was relied on for night-time data only with daytime sleep recorded using paper and pencil as described in previous section. Actigraphy data were processed using AMI’s ActMe Operational Software and the Sadeh Infant Algorithm [35]. Actigraphy data were visually inspected and compared to the sleep diary to reduce artefact with nights that clearly showed unusual readings (e.g., prolonged periods of unexplained zero activity; large discrepancies in bedtime) discarded. The ActMe software categorized the data into the different sleep variables and that were aggregated for each participant. Data analysis was conducted in JASP [36]. In total, 40 families received actigraphs (other families either declined or the device was unavailable). Data from 12 were excluded due to uninterpretable/insufficient data or participant non-compliance (resulting in 0–2 nights of data). The 28 remaining participants included in the final analyses contributed at least four nights (*M* = 7.25 nights, *SD* = 2.41). There were no statistical differences (*p*s > 0.05) between the participants with and without actigraphy data on the measures of family functioning, visual attention and cognitive ability.

The current study included seven sleep variables derived from the actigraphy measure. *Duration*, is the total minutes between the first and last sleep episode of the night. *Sleep Minutes*, is the sum of all epochs scored as sleep. *Wake After Sleep Onset (WASO)* is the sum of epochs scored as awake. *Light Sleep* is the sum of epochs scored as light sleep (classified as asleep but where some movement is detected). *Light Sleep Percentage* is the percentage of non-wake minutes marked as Light Sleep (100 × Light Sleep/(Sleep Minutes + Light Sleep)). *Sleep Efficiency* is the proportion of time marked as sleep (100 × Sleep Minutes/Duration) and *Wake Events* is the count of continuous wake episode blocks.

#### 2.3.2. Visual Attention

Infant visual attention was measured in a lab set up for this purpose. Participants sat on the parent/caregiver’s lap while eye movements were recorded by a Tobii X3-120 eyetracker (Tobii Technology, Stockholm, Sweden) sampling at 120 Hz. Stimuli were presented on a screen placed ~60 cm from the participant using Tobii Pro Studio software. Before the start of each block of trials, participants completed a 5-point calibration.

##### Gap-Overlap

The Gap-Overlap task was adapted from Elsabbagh et al. (2009). At the start of each trial a central stimulus subtending 12° by 12° of visual angle, was presented for 1000 ms. This central stimulus was animated (either jiggling, spinning, or expanding and contracting) for 2000 ms to attract attention to the centre of the screen.

A peripheral target (a cloud) subtending 12° by 14° was presented 13° to the left or right of the central stimuli and remained on the screen until the participant made a fixation on the target, or >3000 ms had elapsed. The target was then replaced by a reward (one of eight animations accompanied by sound), before the next trial. An experimenter observing the participant’s fixations overlaid on a separate screen-controlled stimuli presentation.

Three experimental conditions were presented: Baseline, Gap, and Overlap. The differences between these conditions are in the timings at which the central stimulus disappears, relative to the appearance of the peripheral target. In the Baseline condition, the central stimulus disappears as the peripheral target was presented. In the Gap condition, the central stimulus disappeared 200 ms before the peripheral target, while in the Overlap condition, the central stimulus remained on the screen (as a static image).

Trials were grouped into two blocks of 45 trials. Each of the Gap-Overlap blocks consisted of 15 trials from each condition, presented in random order. The central fixation and the position of the target (left or right of the screen) were counterbalanced. Participants’ response times (RTs) were measured as the time between the appearance of the peripheral target and the participant moving their gaze towards the target. Trials where the participant was not looking at the centre of the screen during target onset, or made a saccade away from the target, were not considered as a valid trial and not included in the analyses.

The two measures of interest were *Disengagement* and *Facilitation*. Disengagement indicates an infant’s ability to avert attention from the central stimulus and orient it towards the newer peripheral target [37]. It was calculated by subtracting RT in the Baseline condition from RT in the Overlap condition. Facilitation measures an infant’s ability to use the disappearance of the central stimulus as a cue to prepare for an upcoming saccade to a peripheral target [37]. It was calculated by subtracting RT in the Gap condition from that of the Baseline condition. Better visual attention is indicated by larger Facilitation and smaller Disengagement.

##### Data Processing of Eye-Tracking Task

Eye-tracking data was processed using the Tobii Pro software. The RT of a trial was defined as the time taken between target onset and the initiation of the saccade towards it. This saccade is identified as the first saccade subtending over 8° (i.e., small saccades within the centre of the screen are ignored). For a trial to be classified as a valid response, the participant’s gaze must have been on the location of the central stimulus when the target appeared. Only responses between 100–1200 ms were considered valid trials [37]. Additionally, a participant’s data for each condition were included only when there were at least two valid trials. This led to the inclusion of data from 40 participants in the final analysis of the Gap-Overlap task.

#### 2.3.3. Family Functioning

The SCORE-28 (Systemic Clinical Outcome and Routine Evaluation) questionnaire [34] was used to measure family functioning. Parents of 42 participants completed the 28-item questionnaire which uses a 6-point Likert scale. The questionnaire measures three subscales of family functioning—Family Strength (e.g., question: “Being in a family is important to us.”); Family Difficulties (e.g., “We find it hard to deal with everyday problems.”); and Communication (e.g., “People in my family interfere too much in each other’s lives”) An overall measure of family functioning was calculated. A higher score on this scale corresponded to greater difficulty functioning within the family context.

#### 2.3.4. Data Analysis

Due to the exploratory nature of the study, where no specific relationships are being tested, the available measures of sleep quality and quantity provided by the actigraphy were considered. Thus, because of the large number of sleep measures, and high multicollinearity within measures, an initial correlation analysis of variables was carried out to help identify cases such as variables that are essentially derivatives of each other (e.g., light sleep percentage and amount of light sleep) before final analysis. While conventional alpha levels of *p* < 0.05 are highlighted, raw (unadjusted) *p*-values are reported as the goal is to identify the sleep measures that have stronger relationships to the visual attention variables.

The influence of sleep and family factors on visual attention, were explored using hierarchical regression which allows for the variables to be entered in a specific order (e.g., to examine the effects of sleep above and beyond that of age and cognition). Separate analyses with Disengagement and Facilitation as outcome variables were conducted. To control for participants’ age and general cognitive abilities, these measures were entered in Step 1, while Light Sleep percentage and DST were entered as the Step 2 input variables. To assess the additional influence of family functioning, the overall family functioning score was entered in Step 3.

For the Gap-overlap task, RTs for a number of the experimental conditions were not normally distributed (Shapiro–Wilk test: *p* < 0.05 for saccades to both cued and uncued locations in Gap and Baseline conditions) the raw RT data for these tasks were log transformed for all analysis as described by Elsabbagh et al. [37]. 

## 3. Results

Table 2 and Table 3 displays the descriptive statistics for the eye-tracking measures and sleep variables, respectively.

### 3.1. Visual Attention

A repeated measures ANOVA showed a statistically significant effect of the three Gap-Overlap conditions, *F*(2,78) = 58.05, *p* < 0.001, *η_p_*^2^ = 0.598. Follow-up paired comparisons showed that both Facilitation (Baseline vs. Gap) and Disengagement (Overlap vs. Baseline) were statistically significant (*t*(39) = 6.41, *p* < 0.001, *d* = 1.01; and *t*(39) = 4.29, *p* < 0.001, *d* = 0.678; respectively). This indicates that the task was able to elicit Facilitation and Disengagement in the current sample.

### 3.2. Sleep Assessment

The initial correlational analysis revealed statistically significant correlations between Disengagement and the sleep measures of Light Sleep percentage and DST, while the only sleep measure that was significantly correlated to Facilitation was DST (see Table 4). Thus, the subsequent regression analysis included Light Sleep percentage along with DST.

### 3.3. Links between Sleep, Family Functioning and Visual Attention

The Step 1 models were not statistically significant for Disengagement (*F*(2,17) = 0.199, *p* = 0.822, *R*^2^*_adj._* = −0.092) or Facilitation (*F*(2,17) = 0.613, *p* = 0.553, *R*^2^*_adj._* = −0.042). The input variables in Step 2 significantly contributed additional explained variance for both Disengagement (∆*R*^2^ = 0.359, *F*(2,15) = 4.35 *p* = 0.032) and Facilitation (∆*R*^2^ = 0.400, *F* (2,15) = 5.64 *p* = 0.015). The overall model with Step 1 and 2 measures was statistically significant for Facilitation only (Disengagement: *F*(4,15) = 2.31, *p* = 0.105, *R*^2^*_adj._* = 0.217; Facilitation: *F*(4,15) = 3.29, *p* = 0.040, *R*^2^*_adj._* = 0.325). The addition of family functioning in Step 3 did not improve either model (Disengagement: ∆*R*^2^ = 0.039, *F*(1,12) = 0.80, *p* = 0.389; Facilitation: ∆*R*^2^ = 0.024, *F*(1,12) = 0.60, *p* = 0.455), with both models statistically non-significant (Disengagement: *F*(5,12) = 1.67, *p* = 0.216, *R*^2^*_adj._* = 0.165; Facilitation: *F*(5,12) = 2.60, *p* = 0.081, *R*^2^*_adj._* = 0.320).

In summary, after controlling for age and cognitive abilities, the sleep measures in Step 2 provided significant relationship to visual attention. Increased DST and lower Light Sleep percentage were associated with better visual attention, as indicated by smaller Disengagement and larger Facilitation. Within both models, DST was statistically significant while Light Sleep percentage was not (see Table 5 for a summary of the hierarchical regression and Figure 1 for visualisations of the relationships between Day Sleep Time, Light Sleep Percentage, Disengagement, and Facilitation).

## 4. Discussion

The primary aim of this exploratory study was to identify potential links between sleep and visual attention in infants. Due to the lack of consensus in defining measures of sleep quality and quantity [30], a secondary goal was to identify which of the measures reported by actigraphy might be useful variables for further research. Finally, we also aimed to explore the potential influence of family functioning on this relation. Results show that more DST and a lower proportion of light sleep at night was associated with better visual attention in infants, with the former being a more robust predictor. Family functioning scores were not related to visual attention.

Observing that longer DST and lower percentage of Light Sleep is related to faster disengagement from the distractor and more efficient orientation to the peripheral target suggests a link between sleep and cognitive development. This result is in line with recent literature linking positive effects of daytime napping to infant cognition. Horger and colleagues [3,38] recently demonstrated how day and night sleep might make independent contributions to infant cognition in their study of motor problem solving. Furthermore, Horváth et al. [39] reported that more frequent daytime naps predicted language development. Lukowski and Milojevich [40] identified DST to positively correlate with immediate and delayed memory recall, while naps after training sessions improved infants’ memory [41] and ability to learn linguistic rules [42,43]. It should be noted that Pisch et al. [7] did not find significant effects of DST on working memory. Although none of these studies looked specifically at attention, they highlight challenges in studying the role of daytime sleep in infants as naps do seem to show positive effects on task performance but is a behaviour that rapidly reduces through early childhood with less habitual napping a marker of typical development.

The current findings are consistent with the only other actigraphy studies of infant cognition in that the quality of night-time sleep, and not its duration, were associated with cognitive performance [7,31,32]. However, it needs to be noted that different sleep quality measures have been identified across studies. For example, Pisch et al. [7] reported significant effects of WASO, but not Wake Events (or DST) while Scher [32] identified Sleep Efficiency and the number of Wake Events as significant predictors. In the present research, Light Sleep percentage indicated significant correlations to measures of visual attention, while the other more frequently reported measures mentioned earlier did not. The negative relation between the percentage of Light Sleep and attention measures does seem to be consistent with our understanding of the effects of later sleep stages (e.g., REM and slow-wave sleep). Research in adults have shown diminished sleep in these stages these to be associated with poorer performance on psychometric tests of cognitive processes such as memory and attention (e.g., [44]). Thus, the measure of light sleep percentage could be capturing the amount of time spent in the latter sleep stages—i.e., more light sleep indicating less latter stage sleep—which is typically more accurately measured using PSG.

The observation of an effect of Light Sleep, but not Sleep Minutes, is particularly notable as research using non-polysomnographic sleep measures typically does not distinguish between these, instead taking all non-wake minutes (i.e., a combination of Sleep Minutes and Light Sleep, or simply the Duration reported in sleep diaries) as a measure of nighttime sleep quantity. The measure of Light Sleep was determined by the Sadeh infant algorithm, which classifies the epochs into one of three states (active/light sleep, quiet sleep, and wake). The algorithm was validated for 1-year-olds wearing the actigraph on the ankle [19]. However, it has been used for studies involving infants up to 36 months (e.g., [45,46]). Sadeh et al. [19] reported 74.9% agreement between classification of light sleep between observers and the algorithm. Thus, it may be useful for future research to go beyond the duration of sleep and also look at the type of sleep being experienced. Furthermore, this highlights the potential use of actigraphy as an alternative to PSG to measure sleep stages in infants (e.g., [47]).

Overall, this study emphasises potential relations between infants’ sleep behaviour and early attention development. While specific measures of sleep quality and quantity were identified as being linked to visual attention, they were not fully consistent with other available research looking at different aspects of infant cognition, highlighting the lack of consensus and understanding of the best indicators of sleep in infancy and childhood [48].

It has been reported that early pathways to later ADHD diagnosis may be observable as early as infancy [49,50]. The Gap-overlap paradigm is a commonly used task used in research on neurodevelopmental disorders such as ADHD (e.g., [51,52,53]), with the typical finding of slower saccadic reactions for children with ADHD. More recently, researchers have also begun to examine the facilitation and disengagement components of this effect and linking them to processes such as baseline arousal and sensitivity to cues (e.g., [54]).

This emphasises the importance of investigating potential relations between early markers of attention control and sleep at a stage where intervention may have its greatest impact. Findings for sleep and attention performance in the current study align with similar findings for children and adults [10,55].

The limitations of this study in relation to sample size are acknowledged, however no other studies have explored the link between objective sleep and concurrent attention control in infants. Notably, only two previous studies have used actigraphy to show relations between infant sleep quality and later attention. Geva, Yaron, and Kuint [6] recently demonstrated a link between poorer preterm infant sleep in an intensive care setting and later attention at four and 18 months. Similarly, Sadeh and colleagues [35] found that sleep at 12 months were related to attention in preschool.

## 5. Conclusions

This study suggests that more daytime sleep (as measured by parental report) and lower proportion of light sleep at night (as measured by actigraphy) are linked to better performance on visual attention tasks in infants, while other sleep measures did not. There was no association with family functioning. It is also important to emphasise the exploratory aspect of the current research which is to identifying specific aspects of sleep that should be more thoroughly examined in future research. In particular, the findings of this study further highlight the emphasis that Gosse and colleagues [2] place on future research in relation to daytime sleep in the context of early cognitive development. These findings are important for the future development of interventions to promote attention development in early childhood. By establishing the potential influence of sleep on attention development in infants, we are gaining an insight into the impact of modifiable factors at a developmental stage where intervention may have the greatest impact.

## Figures and Tables

**Figure 1 children-09-01613-f001:**
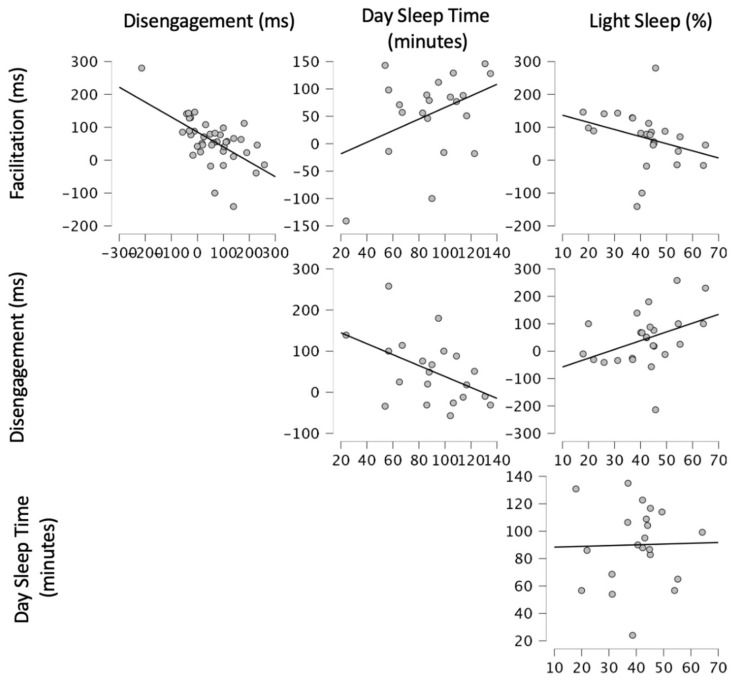
Visualisations of visual attention variables and sleep measures highlighted in the analyses.

**Table 1 children-09-01613-t001:** Number of participants included in the analyses.

Measure	Final Number Included
Actigraphy	28
Sleep Diary	23
Gap-Overlap	40
Score-28 (Family functioning)	42

**Table 2 children-09-01613-t002:** Descriptive statistics (untransformed RT) from the Gap-overlap task.

	Gap (ms)	Baseline (ms)	Overlap (ms)	Facilitation (ms)	Disengagement (ms)
*N*	40	40	40	40	40
*M*	234	293	353	58	61
*SD*	67	63	77	70	97
Minimum	165	216	199	−141	−214
Maximum	509	535	498	280	258

**Table 3 children-09-01613-t003:** Descriptive statistics for sleep variables.

Sleep Measures	*M* (*SD*) [min–max]
Duration (Minutes)	635.37 (72.96) [456.80–740.80]
Sleep (Minutes)	336.79 (87.40) [164.0–461.83]
Total Wake (Minutes)	29.01 (24.51) [1.25–101.0]
Light Sleep (%)	41.49 (11.92) [17.91–64.96]
WASO (Minutes)	29.27 (25.31) [1.25–105.40]
Sleep Efficiency (%)	95.27 (4.74) [77.87–99.82]
Wake Events (Count)	4.64 (2.56) [1–11]
Day Sleep Time (Minutes)	94.32 (37.39) [24.00–211.00]

**Table 4 children-09-01613-t004:** Pearson Correlations of all sleep variables with Facilitation and Disengagement.

		Pearson’s *r* (*p*-Value)						
	Duration	Sleep Minutes	Total Wake	Light Sleep %	WASO	Sleep Efficiency	Wake Events	DST
**Sleep Measures**								
Duration	—							
Sleep Minutes	0.443 *	—						
	(0.018)							
Total Wake	−0.251	−0.563 **	—					
	(0.198)	(0.002)						
Light Sleep %	0.233	−0.707 ***	0.174	—				
	(0.232)	(<0.001)	(0.376)					
WASO	−0.260	−0.564 **	0.997 ***	0.165	—			
	(0.182)	(0.002)	(<0.001)	(0.402)				
Sleep Efficiency	0.384 *	0.578 **	−0.976 ***	−0.124	−0.971 ***	—		
	(0.043)	(0.001)	(<0.001)	(0.531)	(<0.001)			
Wake Events	−0.234	−0.425 *	0.738 ***	0.093	0.738 ***	−0.664 ***	—	
	(0.231)	(0.024)	(<0.001)	(0.636)	(<0.001)	(<0.001)		
Day Sleep Time	−0.176	−0.112	0	0.023	0.006	−0.02	−0.008	—
	(0.445)	(0.630)	(0.998)	(0.923)	(0.098)	(0.930)	(0.974)	
**Family Functioning**	0.035	0.124	−0.39	−0.043	−0.399	0.389	−0.35	−0.023
	(0.870)	(0.563)	(0.060)	(0.843)	(0.053)	(0.060)	(0.094)	(0.923)
**Visual Attention**								
Facilitation	−0.279	0.013	0.284	−0.356	0.303	−0.271	0.287	0.493 *
	(0.178)	(0.950)	(0.169)	(0.080)	(0.141)	(0.190)	(0.163)	(0.023)
Disengagement	0.248	−0.147	−0.202	0.420 *	−0.226	0.178	−0.299	−0.491 *
	(0.232)	(0.484)	(0.334)	(0.037)	(0.277)	(0.394)	(0.146)	(0.024)

* *p* < 0.05, ** *p* < 0.01, *** *p* < 0.001.

**Table 5 children-09-01613-t005:** Hierarchical regression with Disengagement and Facilitation as outcome variables.

		Disengagement			Facilitation		
	Input Variables	*β*	*t*	*p*	*β*	*t*	*p*
Step 1	Age	0.144	0.60	0.557	−0.214	−0.91	0.375
	Cognition	0.035	0.15	0.885	0.165	0.70	0.492
Step 2	Age	0.181	0.88	0.394	−0.240	−1.25	0.229
	Cognition	0.083	0.39	0.702	0.139	0.70	0.493
	Light Sleep %	0.303	1.45	0.167	−0.391	−2.02	0.062
	DST	−0.517 *	−2.46	0.026	0.495 *	2.54	0.023
Step 3	Age	0.235	0.999	0.338	−0.228	−1.075	0.304
	Cognition	0.054	0.230	0.822	0.092	0.436	0.670
	Light Sleep %	0.264	1.143	0.275	−0.427	−2.047	0.063
	DST	−0.607 *	−2.423	0.032	0.456	2.018	0.066
	Family Functioning	−0.227	−0.893	0.389	−0.177	−0.771	0.455

*β*—Standardised regression coefficient, * *p* < 0.05.

## Data Availability

Data from this study are not publicly available as participants consent for this was not requested.

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
