# Peer review of "Links between Daytime Napping, Night-Time Sleep Quality and Infant Attention: An Eye-Tracking, Actigraphy and Parent-Report Study"

_children, 2022, doi:10.3390/children9111613_

Round 1

Reviewer 1 Report

All comments and suggestions can be found in the attached document.

Author Response

We would like to thank the reviewer for their helpful comments. The responses to each query raised are written in blue text below the original comment.

Abstract

  • Use consistent verb tenses
  • Authors could combine the first 2 sentences to say “The current study explored the potential influence of infant sleep, measured by parental report and actigraphy, and family functioning on attention development using eye tracking”. Then the sentence on Line 14 is no longer needed.
  • Line 13 – I would suggest moving the sentence about the sample (starts with Fifty infants…) down to right before the Results

The recommended changes in the Abstract have been made in the manuscript.

Introduction

  • Line 38 – Do authors mean to “…actigraphy as a methodology”? If not, could cut the word “methodology”.

Amended

  • Lines 40-42 - This sentence is not very clear that movement is the basis of actigraphic measurement. Perhaps edit to something more akin to “Actigraphy quantifies the amplitude of movement during sleep and one advantage is that it proves several different measures of….”

Amended

  • Lines 54-56 - Could authors be more explicit in stating if concurrent attention skills are a predictor of later ADHD diagnosis?

Amended and new references added (Line 54 onwards).

  • Line 71 – Section about objective eye tracking methodologies should start it’s own paragraph. It seems the authors highlight this method because they believe the disparities in past research are due to diverging paradigms, but this should be stated outright and used as a transition.

Amended (Line 71 onwards)

  • Lines 78-80 – Description of the Perra et al (2021) study is unclear. Is eye tracking part of both the training and the assessment?

Edited for clarity (Line 91)

  • Line 86 – Unnecessary to open your Current Study section with another summary of the previous research.

Section deleted

  • Line 90 – Consider combining the first 2 paragraphs of this section
  • Line 101-102 – This sentence runs on and could be broken up to read “….cherry-pick sleep variables that are analyzed and reported. As such, the current research explored all available variables provided by the actigraphy measure to report both those that do and do not impact attention”
  • Line 104 – This is really late to introduce one of the primary variables in your analyses. How does family functioning relate to sleep or to attention?
  • This has been amended so that family functioning is introduced earlier (from line 106 in current version) and detail has been expanded.
  • Line 108-118 – Verb tense varies in this paragraph

This section has been re-written in accordance to the reviewers recommendations (Line 97 onwards). Family functioning is introduced earlier (from line 106 in current version) and detail has been expanded.

Methods

  • Line 163 – Was the Action W 2.7 software also used to analyze the data?

The Action W software categorized the data into the different sleep variables and that were aggregated for each participant. Data analysis was conducted in JASP (JASP Team, 2022). This information has now been included in methods section.

  • Line 164 – The usual cut off for the Sadeh Infant algorithm is 12 months. What was the authors justification for continuing to use this algorithm as opposed to the Sadeh algorithm?

An additional line has been added (line 369) which reads: "The algorithm was validated for 1-year-olds wearing the actigraph on the ankle (Acebo et al., 1995). However, it has been used for studies involving infants up to 36 months (e.g., Acebo et al., 1999; 2005)."

  • Line 170 – Did the infants who provided actigraphy data vs. not differ in their measure of family functioning, general cognitive ability, or visual attention?

Thank you for this suggestion. We ran additional analyses and the comparisons between the two groups did not show any group differences. The following line ((177) has been added as well : 

"There were no statistical differences (ps > .05) between the participants with and without actigraphy data on the measures of family functioning, visual attention and cognitive ability."

Table attached below.

Independent Samples T-Test

t

df

p

Family Functioning Total

0.075

40

0.941

Family Strengths

-0.235

40

0.815

Family Difficulties

-0.801

40

0.428

Family Communication

0.834

40

0.409

Age

-1.347

50

0.184

Cognitive abilities 

-0.885

50

0.380

log Facilitation

-1.207

38

0.235

log Disengagement

1.530

38

0.134

IBQ Total

-0.294

26

0.771

  • Line 173 – A subheading or a transition sentence would be beneficial to this paragraph.

The following sentence has been added (Line 180):

"The current study included seven sleep variables derived from the actigraphy measure. Duration.... ."

  • Line 173-179 – There are usually more sleep variables outputted by the Sadeh algorithm (e.g., activity levels, sleep onset latency, longest wake or sleep episode). Why were these excluded?

The initial idea was to explore the variables that are more directly linked to sleep quality and quantity. Variables like mean activity, and the properties of the longest wake and sleep events were thus not included for brevity. While sleep onset latency is a commonly reported variable, we did not include it after feedback from parents that and inspecting the corresponding actigraphy data. SOL is defined as the time between going to bed and the first epoch classified as ‘Sleep’, but there were many instances where ‘going to bed’ did not fit the infants’ sleep routine – e.g., sleep occurring while being rocked in a parent’s arms; in a moving car/buggy; or acti-watches put on only after infant was asleep.

  • Line 181-186 – It would be preferable to include this information in a “Data Analysis Plan” section

A Data Analysis section has been added (Line 251) with line 257 added as well.

  • Line 210 – Please define IOR

Apologies, this is an oversight. A previous version of the manuscript included a description of an additional Inhibition of Return task, which did not prove to be engaging to the participants. All instances of this has been removed.

  • Descriptions of Disengagement and Facilitation and how they can be interpretted were very helpful.

Thank you.

Results

  • Could authors provide a table of the descriptive statistics of infants’ sleep across all variables?

This information is not presented in a new Table 3.

  • Line 243-246 – These sentences could go in a data analysis plan section if added

These lines have been moved.

  • Line 246 – Clarify if the table contains the transformed or raw data

Raw – data was transformed only for the analysis. The heading of Table 2 has  amended to reflect this.

  • Line 256 – Would it be more clear to say “the task was able to elicit Facilitation and Disengagement…”?

Yes, amended.

  • Line 260-263 – Repeated sentences from previously in the manuscript (around Line 181)

Line removed.

  • Line 265 – Why were both light sleep and light sleep percentage included when they are derivative of one another?

We have removed light sleep.

  • Line 295-298 – Move these up to go with the description of the other Steps and how they changed the model

This line has been moved

  • Can Table 3 be consolidated to a single page?

Table (now Table 4) has been reformatted to fit a single page.

  • Authors should consider adding 1-3 figures to display the relationship between significant sleep variables and visual attention measures

A new Figure 1 has been added.

Discussion

  • Line 337-340 - What mechanistic explanation does the significant finding associated with light sleep present?

Information about Light Sleep percentage and how it relates to to later sleep stages and what we know about it’s links to cognitive processes has been added (Lines 353 onwards), which reads: 

"The negative relation between the percentage of Light Sleep and attention measures does seem to be consistent with our understanding of the effects of later sleep stages (e.g., REM and slow-wave sleep). Research in adults have shown diminished sleep in these stages these to be associated with poorer performance on psychometric tests of cognitive processes such as memory and attention (e.g., Kierszenblat & van Swinderen, 2016; Scullin & Bliwise, 2015). Thus, the measure of light sleep percentage could be capturing the amount of time spent in the latter sleep stages - i.e., more light sleep indicating less latter stage sleep – which is typically more accurately measured using PSG."

  • Could the authors unpack some of the trending results from the correlation table? For example, family functioning and wake variables (WASO, total wake, sleep efficiency)

It does seem to suggest a negative relationship between sleep quality and family functioning, in that greater difficulty of family functioning seems to relate to better sleep – higher efficiency, less WASO and less total wake time. However, given the number of correlations we feel that it is not appropriate to attempt to interpret results that do not meet significance in a small population

  • Line 348 – How do Disengagement and Facilitation map onto different criteria for an ADHD diagnosis?

Additional information (lines 382 onwards) has been added which reads:
"The Gap-overlap paradigm is a a commonly used task used in research on neurodevelopmental disorders such as ADHD (e.g., Cairney et al., 2001; Munoz et al., 2003; van der Stigchel et a., 2017), with the typical finding of slower saccadic reactions for children with ADHD. More recently, researchers have also begun to examine the facilitation and disengagement components of this effect, and linking them to processes such as baseline arousal and sensitivity to cues (e.g., Kliberg et.al., 2020)."

  • Authors relate their findings to the rest of the literature well but would benefit from concrete interpretations of each set of results before doing so. For example, the paragraph starting on Line 308 could begin with a summary sentence that longer daytime sleep durations were related to faster times to disengage and to orient to the location of the new stimuli. Essentially, reiterate the points stated in the Results section, Lines 288-290.

An additional sentence has been added (Line 331): 

"Observing that longer DST and lower percentage of Light Sleep is related to faster disengagement from the distractor and more efficient orientation to the peripheral target suggests a link between sleep and cognitive development." 

Minor comments

  • Extra parenthesis on pg 1 line 37

Thank you for spotting this.

Reviewer 2 Report

This work showed an investigation about the potential influence of infant sleep on attention development using parental report, sleep actigraphy and eye-tracking. The topic of the work is of interest, but some parts were not well described.  However, I have a number of further comments to improve the manuscript:

Introduction:

·     It’s better to mention and describe the statistical (analysis) method that used in this work. Then, explain why this work used it based on the literature review.

·    It’s better to describe the state-of-the-art of this work to ensure the novelty of this work.

Material and Methods:

·    How long the sleep duration (or days) of each subject that used in this work?

·    What kind of the questionnaires that used in this work? Is it any reason to use it?

·    What is RT? it’s better write it down first before abbreviation it.

·    Why the author distinguishes the measuring process: daytime sleep duration measured by parental report, and proportion of light sleep at night measured by actigraphy? I think both of them can be measured by parental report and actigraphy, thus it will validate the results.

Results and discussion:

·    Table 3 and related argument: How authors have selected only these sleep measures variables such as, Total Wake, Light Sleep, WASO, Sleep Efficiency, Wake Events, and DST. It should be commented on in the manuscript since there are a lot of sleep quality variables.

·    How we can know that a higher proportion of light sleep, a marker of poorer sleep quality, and less daytime sleep, were negatively linked with facilitation and disengagement on the Gap-Overlap task ?

References:
Are there any more updated references? Please use references at most five years!

Author Response

We would like to thank the reviewer for their helpful comments. The responses to each query raised are written in blue text below the original comment.

Introduction:

  • It’s better to mention and describe the statistical (analysis) method that used in this work. Then, explain why this work used it based on the literature review.

A Data Analysis section has been added (Line 251)

  • It’s better to describe the state-of-the-art of this work to ensure the novelty of this work.

The introduction has been amended to reflect more recent literature and highlight the novel contributions.
Material and Methods:

  • How long the sleep duration (or days) of each subject that used in this work?

This information is now on Line 177.

  • What kind of the questionnaires that used in this work? Is it any reason to use it?

Additional information about the questionnaire has been added to the section (Line 240 onwards). Additional information about the links of family functioning has also been added (Line 86 onwards).

  • What is RT? it’s better write it down first before abbreviation it.

Apologies for missing missing this -added to line 215

  • Why the author distinguishes the measuring process: daytime sleep duration measured by parental report, and proportion of light sleep at night measured by actigraphy? I think both of them can be measured by parental report and actigraphy, thus it will validate the results.

Participants were requested to wear the actigraphy watches for night sleep as some initial parents  indicated difficulty in getting infants to comply during the day (this is noted on line 182-3). The entries from the sleep diary was indeed used to corroborate the actigraphy data

Results and discussion:

·    Table 3 and related argument: How authors have selected only these sleep measures variables such as, Total Wake, Light Sleep, WASO, Sleep Efficiency, Wake Events, and DST. It should be commented on in the manuscript since there are a lot of sleep quality variables.

The exploratory nature of the research and analysis has been highlighted in the Data Analysis section (Line 251).

  • How we can know that a higher proportion of light sleep, a marker of poorer sleep quality, and less daytime sleep, were negatively linked with facilitation and disengagement on the Gap-Overlap task ?

The effects of facilitation and disengagement are in different directions. Lines 221 onwards explains the two measures. The direction of relationship can be ascertained in the tables, with negative numbers indicating a negative effect.

References:
Are there any more updated references? Please use references at most five years!

Additional references have been added.

Round 2

Reviewer 2 Report

The manuscript has been well revised by the author.